# Performance Improvement of Discretely Modulated Continuous-Variable Quantum Key Distribution with Untrusted Source via Heralded Hybrid Linear Amplifier

**DOI:** 10.3390/e22080882

**Published:** 2020-08-12

**Authors:** Kunlin Zhou, Xuelin Wu, Yun Mao, Zhiya Chen, Qin Liao, Ying Guo

**Affiliations:** 1School of Traffic and Transportation Engineering, Central South University, Changsha 410083, China; yingguo1001@foxmail.com (K.Z.); chzy@csu.edu.cn (Z.C.); 2Jiangsu Key Construction Laboratory of IoT Application Technology, Taihu University, Wuxi 214064, China; 3School of Automation, Central South University, Changsha 410083, China; 4College of Computer Science and Electronic Engineering, Hunan University, Changsha 410082, China; llqqlq@hnu.edu.cn

**Keywords:** untrusted source, discrete modulation, heralded hybrid linear amplifier

## Abstract

In practical quantum communication networks, the scheme of continuous-variable quantum key distribution (CVQKD) faces a challenge that the entangled source is controlled by a malicious eavesdropper, and although it still can generate a positive key rate and security, its performance needs to be improved, especially in secret key rate and maximum transmission distance. In this paper, we proposed a method based on the four-state discrete modulation and a heralded hybrid linear amplifier to enhance the performance of CVQKD where the entangled source originates from malicious eavesdropper. The four-state CVQKD encodes information by nonorthogonal coherent states in phase space. It has better transmission distance than Gaussian modulation counterpart, especially at low signal-to-noise ratio (SNR). Moreover, the hybrid linear amplifier concatenates a deterministic linear amplifier (DLA) and a noiseless linear amplifier (NLA), which can improve the probability of amplification success and reduce the noise penalty caused by the measurement. Furthermore, the hybrid linear amplifier can raise the SNR of CVQKD and tune between two types of performance for high-gain mode and high noise-reduction mode, therefore it can extend the maximal transmission distance while the entangled source is untrusted.

## 1. Introduction

Quantum key distribution (QKD) allows two legitimate parties to share the secure key string over an insure quantum channel [1,2,3,4,5]. The continuous-variable QKD (CVQKD) is one of alternative protocol of QKD, can provide higher detection efficiencies than original discrete-variable QKD (DVQKD) [6,7,8,9,10,11,12]. Furthermore, using the lase to generate the entangled source makes it easier to integrate with existing fiber network systems. However, compared with the DVQKD, the CVQKD has one of fatal disadvantages, which is the short secure transmission distance [11,13,14], especially if the entangled source is untrusted [15]. In fact the safety of CVQKD protocol has been proved where the entangle source is originated from the malicious eavesdropper [15], and it still can distill a positive key rate. However, the performance of this protocol is not great in terms of secret key rate and transmission distance, and thus could restrict the practical application of CVQKD in development. One of the main reasons for the short transmission distance of CVQKD is that traditional Gaussian modulation CVQKD cannot maintain relatively high reconciliation efficiency in long-distance transmission, but discretely the modulation scheme can solve this problem even better than using error correcting code [16]. The four-state discretely modulation scheme produces four nonorthogonal coherent states in phase space and thus it can randomly select those coherent states by different quadrature to encode information rather than using the $\hat{x}$ and $\hat{p}$ quadratures themselves. Due to the fact that, the received coherent state will be easier to distinguish with a traditional Gaussian counterpart. That is the reason why the discrete modulation scheme can reach relative longer transmission distance even at very low SNR [16].

The penalty noise caused by detector in Bob side will also directly affect the transmission distance performance of CVQKD protocol where the entanglement source is untrusted. In practice, the intrinsic disadvantage of the receiver’s apparatus, such as the efficiency of detector and the electrical noise referred by measurement, will decrease the secret key rate and thus shorten the transmission distance. In fact, optical amplifiers can solve such problem, for instance, the noiseless linear amplifier (NLA) is one of existing methods to overcome this limitation of detector. It can compensate the noise penalty and thus preserve the signal-to-noise ratio (SNR) [17,18,19,20,21]. However, the CVQKD scheme with non-traditional light source modulation and non-Gaussian operation, such as the discrete or unimensional modulation and photon subtraction operation, cannot directly implement via NLA. To overcome this problem, the measurement-based NLA (MB-NLA) has been proposed [22,23], which installs a dual homodyne detection ahead of NLA. The MB-NLA not only solve the application problem of NLA in CVQKD shceme with non-Gaussian modulation, but also can simulate the action of post-selection with amplifier. Furthermore, the quantum filter induced by the MB-NLA can simply model an appropriate postprocessing [24,25]. Although MB-NLA has the advantages mentioned above, its property of post-selective restricts it in point-to-pint application such as CVQKD. To overcome this defect, the hybrid amplifier has been proposed, which concatenates a deterministic liner amplifier (DLA) behind the MB-NLA to form a feed-forward loop [25]. Due to the installation of the DLA, the amplification function uncertainty caused by NLA has been solved [24]. Furthermore, the DLA also can compensate the degeneration of SNR caused by NLA. It is worth noting that DLA can be divided into a phase-sensitive amplifier (PSA) and phase-insensitive amplifier (PIA) [26,27]. If Bob adopts heterodyne measurement, we only can deploy PIA as the DLA, on the contrary, it can deploy PSA [28,29].

From the inspiration of the discrete modulation and hybrid amplifier mentioned above, in this paper, we proposed a scheme of discretely modulated CVQKD with hybrid amplifier to improve the performance of CVQKD with untested entangled source. Here, the four-state discrete modulation can improve the transmission distance of CVQKD even on the low SNR. Moreover, a hybrid amplifier deployed on the Bob side can reduce the influence of electrical noise caused by the detection and thus further enhance the transmission distance and secret key rate. The hybrid amplifier not only integrates the advantages of MB-NLA and DLA, but also provides a fully tunable gain for NLA and DLA. The proposed scheme outperforms the classical Gaussian modulation CVQKD with untested source in terms of the key rate, and also transcends the traditional discrete modulation CVQKD with untested source in terms of the maximal transmission distance.

This paper is structured as follows. In Section 2, we introduce the scheme design for four-sate discrete modulation CVQKD with untested source involving the hybrid amplifier. In Section 3, we demonstrate the secret key rate for the proposed scheme. In Section 4, we analyze the results of its performance. In Section 5, we provide a conclusion.

## 2. Discretely Modulated CVQKD with Untested Source via Hybrid Amplifier

In this paper, we consider to integrate a hybrid amplifier at Bob side in discretely modulated CVQKD scheme with untested source and thus leading to the performance improvement of transmission distance at low SNR. In this section, we will introduce the Alice side, Eve side, and Bob side according to the order from left to right in Figure 1, namely, the four-state discrete modulation will be introduced at first, next we consider about the CVQKD scheme with untested source model, and the last part is the description of hybrid amplifier.

### 2.1. Deploying a Four-State Discrete Modulation at Alice Side

We first begin by introducing the four-state discrete modulation scheme. The four-state protocol is belonged to the discrete modulation protocol. In the discrete modulation CVQKD, Alice produces *N* coherent states at phase space with
(1)$$|\alpha_{k}^{N}\rangle =|\alpha e^{\frac{i2k\pi }{N}}\rangle .$$

For the four-state protocol, we only prepare four coherent states at phase space (see the Figure 2) [12], namely, here we have $|\alpha_{k}^{N}\rangle =|\alpha e^{\frac{i(2k+1)\pi }{4}}\rangle$, k∈{0,1,2,3}, and α is a positive parameter and relates to the modulation variance with $\alpha =\sqrt{\frac{V_{M}}{2}}$.

In the scheme of prepare-and-measure (PM), Alice randomly selects one of coherent state $|\alpha_{k}^{4}\rangle ,k\in \left\{0,1,2,3\right\}$, and transmits it to Bob by quantum channel with transmission efficiency *T* and excess noise ϵ. Bob use detector to measure the received coherent states with detection efficiency $T_{d}$ and electronics noise $v_{el}$. The mixture of four coherent state received by Bob can be denoted as
(2)$$\rho_{4c}=\frac{1}{4}\sum_{k=0}^{3}|\alpha_{k}^{4}\rangle \langle \alpha_{k}^{4}|.$$

At the equivalent scheme of entanglement-based (EB), we can more convenient to calculate the secret key rate. First, the $\rho_{4c}$ is transformed to
(3)$$\rho_{4c}=tr_{A}(|\Phi \rangle \langle \Phi |),$$
the mixture coherent state can be diagonalized and thus rewritten as the following form,
(4)$$\rho_{4c}=\mu_{0}|\phi_{0}\rangle \langle \phi_{0}|+\mu_{1}|\phi_{1}\rangle \langle \phi_{1}|+\mu_{2}|\phi_{2}\rangle \langle \phi_{2}|+\mu_{3}|\phi_{3}\rangle \langle \phi_{3}|,$$
here
(5)$$\mu_{0,2}=\frac{1}{2}e^{-\alpha^{2}}\left[cosh\left(\alpha^{2}\right)\pm cos\left(\alpha^{2}\right)\right],$$
(6)$$\mu_{1,3}=\frac{1}{2}e^{-\alpha^{2}}\left[sinh\left(\alpha^{2}\right)\pm sin\left(\alpha^{2}\right)\right],$$
and
(7)$$|\phi_{k}\rangle =\frac{e^{-\frac{\alpha^{2}}{2}}}{\sqrt{\mu_{k}}}\sum_{n=0}^{\infty }\frac{\alpha^{4n+k}}{\sqrt{(4n+k)!}}{\left(-1\right)}^{n}|4n+k\rangle .$$

Then, we purify the system by the Schmidt decomposition
(8)$$|\Phi \rangle =\frac{1}{2}\sum_{k=0}^{3}|\psi_{k}\rangle |\alpha_{k}\rangle ,$$
where the non-Gaussian (NG) state $|\psi_{k}\rangle$ can be expressed as
(9)$$|\psi_{k}\rangle =\frac{1}{2}\sum_{m=0}^{3}e^{-i\left(1+2k\right)m\left(\frac{\pi }{4}\right)}|\phi_{m}\rangle .$$

### 2.2. Eve Producing the Untrusted Entanglement Source

In practice, we need to consider any possible scenarios like that Eve that could have controlled the entangled source and generate it to any state. In this case, we first assume that Eve adopts Gaussian modulation and compare its performance to discrete modulation counterpart. That is because the Gaussian state can produce the maximum Shannon mutual information. In Gaussian attack, Eve perfectly uses its own quantum channel to replace the quantum channel which is placed between Alice and Bob. The loss channels are simulated by two independent beam splitters with transmissions $T_{1}$ and $T_{2}$ [12]. Note that if there is a symmetric transmission ($T_{1}=T_{2}$), it can be considered as entangled source in middle (ESIM) CVQKD protocol, and if these two transmissions are asymmetric with $T_{1}=1$ ($T_{1}\neq T_{2}$), it can be recovered to the traditional CVQKD protocol where entangled source is generated at Alice or Charlie. In this case, we hypothesize entangled source is generated by Eve, but Eve is close to Alice side. Here, we have the distance between Alice and Eve with $L_{1}$ and Eve between Bob with $L_{2}$, therefore the channels transmissions can be rewritten as $T_{1}=10^{-0.02L_{1}}$, $T_{2}=10^{-0.02L_{2}}$, and thus $T=T_{1}T_{2}$.

Eve’s Einstein–Podolsky–Rosen (EPR) state ${|\psi \rangle }_{AB}$ is created by two single-mode squeezed states, |z〉 and |−z〉 with
(10)$${|z\rangle }_{i}={\hat{O}}_{i}(z|0\rangle ,$$
here ${\hat{O}}_{i}\left(z\right)$ is the squeezing operator and can be expressed with
(11)$${\hat{O}}_{i}\left(z\right)=exp\left[-\frac{z}{2}\left({\hat{a}}_{i}^{†2}-{\hat{a}}_{i}^{2}\right)\right],$$
where ${\hat{a}}_{i}$ and ${\hat{a}}_{i}^{†}$ represent the creation and annihilation operation, and *z* is the squeezing parameter. The combining two single-mode squeezed states (${\hat{z}}_{A_{0}}$ and ${\hat{z}}_{B_{0}}$) become a EPR state ${|\psi \rangle }_{AB}$ by a 50:50 beam splitter. We can denote each entangled mode (${\hat{X}}_{A}$ and ${\hat{X}}_{B}$) as
(12)$${\hat{X}}_{A}=\frac{\left({\hat{z}}_{A_{0}}+{\hat{z}}_{B_{0}}\right)}{\sqrt{2}},$$
(13)$${\hat{X}}_{B}=\frac{\left({\hat{z}}_{A_{0}}-{\hat{z}}_{B_{0}}\right)}{\sqrt{2}}.$$

The EPR state ${|\psi \rangle }_{AB}$ can be represent as
(14)$$|\psi \rangle ={\hat{O}}_{AB}{\left(-z\right)|0\rangle }_{A}{|0\rangle }_{B}=\delta \sum_{n=0}^{\infty }\lambda^{n}|n,n\rangle ,$$
where ${\hat{O}}_{AB}$ represents a squeezing operator on two modes (modes *A* and *B*), ${\hat{O}}_{AB}\left(z\right)=exp\left[-z\left({\hat{a}}_{A}^{†}{\hat{a}}_{B}^{†}-{\hat{a}}_{A}{\hat{a}}_{B}\right)\right]$, $\lambda =\sqrt{\frac{V-1}{V+1}}$, here *V* is variance of two modes and $\delta =\sqrt{1-\lambda^{2}}$. Then, we assume that Eve performs the best collective Gaussian attack on the entangled source modes. The entangling cloner, a common collective Gaussian attack, is used to prepare two ancilla modes, ${\hat{X}}_{E_{1}}$ and ${\hat{X}}_{E_{2}}$, from an entangled Gaussian state with symmetrized variance ($W_{1}=W_{2}$). In each pulse, Eve stores modes $E_{11}$ and $E_{21}$ in her quantum memory and injects the other two modes $E_{12}$ and $E_{22}$ into the beam splitter, thus obtains the output modes $E_{13}$ and $E_{23}$. In the end of protocol, Eve measures the quadratures of $E_{11}$ and $E_{21}$ to obtain the communication information between Alice and Bob.

### 2.3. Implementing a Hybrid Linear Amplifier at Bob Side

First, we introduce the conceptual layout of our scheme, and in order to overcome the post-selective nature and thus promote the point-to-point application for traditional linear amplifier, we set a feed-forward loop outputted to a quantum state instead of the original state as follows [24],
(15)$${\hat{\rho }}_{out}=ZTr_{v}\{{\hat{D}}_{g_{D}}g_{N}^{\hat{n}}{\hat{\rho }}_{in}⊗|0\rangle \langle 0{|}_{v}g_{N}^{\hat{n}}{\hat{D}}_{g_{D}}^{†}\}.$$
where the *Z* is the normalization factor and the operators ${\hat{D}}_{g_{D}}$ and $g_{N}^{\hat{n}}$ are used to model the action of NLA and DLA. In addition, the input coherent state can be written as
(16)$${\hat{\rho }}_{in}=\frac{1}{\pi }\frac{1-\lambda^{2}}{\lambda^{2}}\int d^{2}\alpha e^{-\frac{1-\lambda^{2}}{\lambda^{2}}{\left|\alpha \right|}^{2}}|\alpha \rangle \langle \alpha |.$$
here the λ (0≤λ<1) relates to the variance of coherent states with *V*. Due to the operations of NLA and DLA, the variance has been changed to
(17)$$V=\frac{1+\lambda^{2}}{1-\lambda^{2}}⟹V_{g}=\frac{1+g_{N}^{2}\lambda^{2}}{1-g_{N}^{2}\lambda^{2}}.$$

The mean variance of two quadratures of the electric field can be written as
(18)$$\langle \hat{X}\rangle =\hat{a}+{\hat{a}}^{†},$$
(19)$$\langle \hat{P}\rangle =-i\left(\hat{a}-{\hat{a}}^{†}\right).$$
when the signal feed through the hybrid amplifier, the expectation of the measurable value $\hat{M}=\left(\hat{a},{\hat{a}}^{†}\right)$ is further amplified by DLA, and the outcomes can be expressed as
(20)$${\hat{a}}_{out}={\hat{a}}_{in}g_{D}+{\hat{a}}_{int}^{†}\sqrt{g_{D}^{2}-1},$$
(21)$${\hat{a}}_{out}^{†}={\hat{a}}_{in}^{†}g_{D}+{\hat{a}}_{int}\sqrt{g_{D}^{2}-1}.$$

So that, according to above Equations (Equation 18)–(Equation 21), we can give the corresponding outcome quadratures of amplitude and phase as
(22)$${\langle \hat{X}\rangle }_{out}={\langle \hat{X}\rangle }_{in}g_{N}g_{D},$$
(23)$${\langle \hat{P}\rangle }_{out}={\langle \hat{P}\rangle }_{in}g_{N}g_{D}.$$

In this proposed scheme, we take the GG02 protocol [2] (the GG02 protocol is a fundamental CVQKD protocol) as usual but add a hybrid amplifier at Bob side, so that the secure key rate should depend on the covariance matrix with presence of the hybrid amplifier. However, the output of the linear hybrid amplifier remains in the Gaussian regime, therefore we use an equivalent channel to replace the previous one (Figure 3). Here, the corresponding system parameters are changed from ($|\lambda \rangle ,T_{1},T_{2},\epsilon ,\chi$) to ($|\zeta \rangle ,\eta_{1},\eta_{2},\epsilon_{gn},\chi_{gd}$), those equivalent parameters are listed below,
(24)$$\zeta =\lambda \sqrt{\frac{\left(g_{N}^{2}-1\right)\left(\epsilon -2\right)T-2}{(g_{N}^{2}-1)\epsilon T-2}},$$
(25)$$\begin{array}{c} \eta =\frac{g_{N}^{2}T}{\left(g_{N}^{2}-1\right)T\left[\frac{1}{4}\left(g_{N}^{2}-1\right)\left(\epsilon -2\right)\epsilon T-\epsilon +1\right]+1}, \end{array}$$
(26)$$\epsilon_{g_{N}}=\epsilon -\frac{1}{2}\left(g_{N}^{2}-1\right)\left(\epsilon -2\right)\epsilon T,$$
(27)$$\chi_{hom}^{g_{D}}=\frac{\left(1-T_{d}\right)+v_{el}}{g_{D}T_{d}},$$
and
(28)$$\chi_{het}^{g_{D}}=\frac{1+\left(1-T_{d}\right)+2v_{e}+\left(g_{D}-1\right)T_{d}}{g_{D}T_{d}}.$$

Finally, we present the equivalent experimental layout for our scheme. As mentioned above, we consider applying a hybrid linear amplifier at Bob’s station (as shown in Figure 4). The input mode $B_{1}$ is first fed through a beam splitter with transmissivity $T_{g}={\left(g_{N}/g_{D}\right)}^{2}$, here $g_{N}$ and $g_{D}$ represent gains of NLA and DLA, respectively. After that, the reflected mode is fed through a measurement-based noiseless linear amplifier (MB-NLA), which consists by a dual-heterodyne detection and a noiseless linear amplifier. Here, one of the heterodyne detectors is used to measure the *X* quadrature of the coherent state, and other one measures the *P* quadrature. The MB-NLA has some special features, including a probabilistic of Gaussian filter and rescaling of amplifier gain factors. Here, the rescaling of gain NLA is given as $g_{N}^{\prime }=\frac{1}{g_{N}}$, and thus the success probability of Gaussian filter is given as [25,30]
(29)$$P\left(\alpha_{m}\right)=\left\{\begin{array}{c} exp(|\alpha_{m}{|}^{2}-|\alpha_{c}{|}^{2})\left[1-{\left(g_{N}\right)}^{2}\right],\;\left|\alpha_{m}\right|\leq \alpha_{c}\\ 1.\;\;\;|\alpha_{m}|>\alpha_{c} \end{array}\right.$$

The measurement outcomes of the dual-heterodyne detection is applied as [23]
(30)$$\alpha_{m}=\frac{x_{m}+ip_{m}}{\sqrt{2}},$$
and $\alpha_{c}$ represents a tunable cut-off parameter, which directly determines the success probability of the protocol and how closely the MB-NLA approximates to an ideal NLA ($\alpha_{c}>0$). In this scheme, we couple a DLA before the MB-NLA to further improve the probability of success by $g_{D}^{\prime }=\sqrt{2(g_{D}^{2}-1)}$. Finally, the output signal feeds through an elector-optic modulator (EOM) and a beam splitter with transmissivity 99:1.

Notice that the performance of our hybrid amplifier depends on the respective gain of two different amplifiers. A relatively larger NLA gain cloud increases the signal-to-noise ratio (SNR), but also leads to a lower success probability. In contrast, a relatively larger DLA gain would increase the success probability, but will also bring the added noise.

## 3. Simulation of the Secret Key Rate

In this section, we will demonstrate the calculation process of the asymptotic secret key rate for traditional Gaussian modulation and discrete modulation. The traditional Gaussian modulation scheme is based on two quantum states (coherent or squeezed states) and two measurement methods (homodyne or heterodyne detection). The discrete modulation scheme is based on coherent state four-state modulation and homodyne detection.

### 3.1. The Gaussian Modulation with Untested Source via Hybrid Amplifier Scheme

In the Gaussian modulation scheme, we will mainly introduce calculation of direct reconciliation, and for the reverse reconciliation the calculation process can be simply derived form the covariance matrix $\Gamma_{A_{2}B_{2}}$ by switching *a* and *b*. The secret key rate is defined as [2]
(31)$$K=\beta I\left(A_{2}:B_{3}\right)-\chi_{E},$$
here β represents the reconciliation efficiency and $I(A_{2}:B_{3})$ is the Shannon mutual information between Alice and Bob [31]; furthermore, the $\chi_{E}$ is Eve’s mutual information connected between Eve and Alice for direct reconciliation or between Eve and Bob for reverse reconciliation. When we consider the situation of untested entangled source CVQKD protocol with hybrid amplifier at Bob side, the covariance matrix of the Gaussian state $\zeta_{A_{2}B_{3}}$ is given by (notice that here we assume that Alice and Bob both have deployed the ideal detectors)
$$\begin{array}{c} \Gamma_{A_{2}B_{3}}=\left(\begin{array}{cc} aI & c\sigma_{z}\\ c\sigma_{z} & bI \end{array}\right), \end{array}$$
where *I* and $\sigma_{z}$ are the Pauli matrices $a=\eta_{1}V+\left(1-\eta_{1}\right)W_{1}$, $b=\eta_{2}V+\left(1-\eta_{2}\right)W_{2}$, and $c=\sqrt{\eta_{1}}\sqrt{\eta_{2}}\sqrt{V^{2}-1}$; therefore, the covariance matrix $\Gamma_{A_{2}B_{3}}$ can be rewritten as
$$\begin{array}{c} \left(\begin{array}{cc} \eta_{1}V+\left(1-\eta_{1}\right)W_{1}I & \sqrt{\eta_{1}}\sqrt{\eta_{2}}\sqrt{V^{2}-1}\sigma_{z}\\ \sqrt{\eta_{1}}\sqrt{\eta_{2}}\sqrt{V^{2}-1}\sigma_{z} & \eta_{2}V+\left(1-\eta_{2}\right)W_{2}I \end{array}\right), \end{array}$$
here $W_{i}=\eta_{i}\chi_{line}/\left(1-\eta_{i}\right)\left(i=1,2\right)$ and $\chi_{line}=\left(1-\eta_{i}\right)/\left(\eta_{i}+\epsilon_{gn}\right)$ represents the added noise inputted by Gaussian channel. We first introduce the case of squeezed states with homodyne or heterodyne detection. In this case, when Bob adopts homodyne measurement, the mutual information between Alice and Bob can be defined as [2]
(32)$$I{\left(A_{2}:B_{3}\right)}_{hom}^{S}=\frac{1}{2}log_{2}\left(\frac{a}{a-c^{2}/b}\right),$$
with heterodyne measurement the mutual information is given as
(33)$$I{\left(A_{2}:B_{3}\right)}_{het}^{S}=\frac{1}{2}log_{2}\left(\frac{a}{a-(c^{2}/\left(b+1\right))}\right).$$

Then, we will introduce the Eve’s mutual information, namely, the $\chi_{E}$ mentioned in Equation (Equation 31). Here,
(34)$$\chi_{E}=S\left(E\right)-S\left(E|A\right),$$
and it has the same calculation process for both homodyne and heterodyne detections. Due to the direction reconciliation implemented in this scheme, which only depends on the measurement results with Alice but not Bob. Furthermore, Eve provides a purification for Alice and Bob’s matrix, so that we can rewrite S(E) as
(35)$$S\left(AB\right)=G\left[\left(\lambda_{1}-1\right)/2\right]+G\left[\left(\lambda_{2}-1\right)/2\right].$$

Here, $G\left(x\right)=\left(x+1\right)log_{2}\left(x+1\right)-xlog_{2}x$, thus the symplectic eigenvalues are represented as
(36)$$\lambda_{1,2}^{2}=\frac{1}{2}\left[A\pm \sqrt{A^{2}-4B^{2}}\right].$$
where $A=a^{2}+b^{2}-2c^{2}$ and $B=ab-c^{2}$, then Eve uses the same purification for Alice and Bob’s matrix, thus we can rewrite S(E|A) as S(B|A). Here, $S\left(B|A\right)=G[\left(\lambda_{3}-1\right)/2]$ and $\lambda_{3}=\sqrt{b(b-\left(c^{2}/a\right))}$. Therefore, we now can calculate the secret key rate mentioned in Equation (Equation 31).

Next, when Alice adopts coherent state, the mutual information for Alice and Bob with homodyne and heterodyne detections are given by
(37)$$I{\left(A_{2}:B_{3}\right)}_{hom}^{C}=\frac{1}{2}log_{2}\left(\frac{a+1}{a+1-c^{2}/b}\right),$$
(38)$$I{\left(A_{2}:B_{3}\right)}_{het}^{C}=log_{2}\left(\frac{b+1}{b+1-c^{2}/\left(a+1\right)}\right).$$
in the case of coherent state with homodyne detection and heterodyne detection, the S(E) mentioned in Equation (Equation 34) is same with the counterpart of squeezed state, but the S(E|A) is replaced by S(BC|A) and expressed as follows,
(39)$$S\left(BC|A\right)=G\left[\left(\lambda_{4}-1\right)/2\right]+G\left[\left(\lambda_{5}-1\right)/2\right],$$
here
(40)$$\lambda_{4,5}^{2}=\frac{1}{2}\left[C_{hom}\pm \sqrt{C_{hom}^{2}-4D_{hom}}\right].$$
where, in the case of homodyne detection, the $C_{hom}$ and $D_{hom}$ can be denoted as
(41)$$C_{hom}=\frac{A\chi_{hom}^{g_{D}}+V\sqrt{B}+\eta \left(V+\chi_{line}\right)}{\eta (V+\chi_{tot})},$$
(42)$$D_{hom}=\sqrt{B}\frac{V+\sqrt{B}\chi_{hom}^{g_{D}}}{\eta (V+\chi_{tot})}.$$

For the heterodyne case, the symplectic eigenvalues 6 and 7 can be expressed as
(43)$$\lambda_{6,7}^{2}=\frac{1}{2}\left[C_{het}\pm \sqrt{C_{het}^{2}-4D_{het}}\right],$$
and $C_{het}$ and $D_{het}$ are denoted as follows,
(44)$$C_{het}=\frac{A{\left(\chi_{het}^{g_{D}}\right)}^{2}+B+1+2\chi_{het}^{g_{D}}\left[V\sqrt{B}+\eta \left(V+\chi_{line}\right)\right]+2\eta \left(V^{2}-1\right)}{{\left[\eta \left(V+\chi_{tot}\right)\right]}^{2}},$$
(45)$$D_{het}={\left(\frac{V+\sqrt{B}\chi_{het}^{g_{D}}}{\eta (V+\chi_{tot})}\right)}^{2}.$$

$\chi_{line}=1/\eta -1+\varepsilon$ is shot noise units and $\chi_{tot}=\chi_{line}+\chi_{h}/\eta$ is channel total noise, here $\chi_{h}$ is detection added noise mentioned in Equations (Equation 27) and (Equation 28) with $\chi_{hom}^{g_{D}}$ and $\chi_{het}^{g_{D}}$. We can now plot the final secret key rate for both the Gaussian modulation squeezed state and coherent state.

### 3.2. The Discrete Modulation with Untested Source via Hybrid Amplifier Scheme

In the discrete modulation scheme, we mainly consider direct reconciliation with the coherent state scheme. The secret key rate is also same as Equation (Equation 31), but due to the state $\zeta_{A_{2}B_{3}}^{DM}$, is not Gaussian anymore, and we should replace the Gaussian state $\zeta_{A_{2}B_{3}}$ to $\zeta_{A_{2}B_{3}}^{DM}$ as follows,
$$\begin{array}{c} \Gamma_{A_{2}B_{3}}^{DM}=\left(\begin{array}{cc} a_{DM}I & c_{DM}\sigma_{z}\\ c_{DM}\sigma_{z} & b_{DM}I \end{array}\right), \end{array}$$
here $a_{DM}=\eta_{1}a_{0}+\left(1-\eta_{1}\right)W_{1}$, $b_{DM}=\eta_{2}b_{o}+\left(1-\eta_{2}\right)W_{2}$, and $c_{DM}=\sqrt{\eta }c_{0}$, the $a_{0}$,$b_{0}$ and $c_{0}$ can express as following
(46)$$a_{0}=1+2\alpha^{2},$$
(47)$$b_{0}=1+2\alpha^{2},$$
(48)$$c_{0}=2\alpha^{2}\sum_{k=0}^{3}\mu_{k-1}^{3/2}\mu_{k}^{-1/2}.$$
where $\alpha =\sqrt{\frac{V_{M}}{2}}$ and $V_{M}=V-1$; the mutual information between Alice and Bob is also the same with Gaussian counterpart in Equation (Equation 37), but the $\chi_{E}$ has been changed, $\chi_{E}=S\left(E\right)-S\left(E|A\right)$, here the S(E) can be denoted as
(49)$$S\left(E\right)=S\left(AB\right)=G\left[\left(\lambda_{1}^{DM}-1\right)/2\right]+G\left[\left(\lambda_{2}^{DM}-1\right)/2\right].$$

Here $G\left(x\right)=\left(x+1\right)log_{2}\left(x+1\right)-xlog_{2}x$, thus the symplectic eigenvalues are represented as
(50)$$\lambda_{1,2}^{DM}=\sqrt{\frac{1}{2}\left(A_{DM}\pm \sqrt{A_{DM}^{2}-4B_{DM}^{2}}\right)},$$
with $A_{DM}=a_{DM}^{2}+b_{DM}^{2}-2c_{DM}^{2}$ and $B_{DM}=a_{DM}b_{DM}-c_{DM}^{2}$, furthermore the S(E|A) is also replaced by S(BC|A)
(51)$$S\left(BC|A\right)=G\left[\left(\lambda_{3}^{DM}-1\right)/2\right]+G\left[\left(\lambda_{4}^{DM}-1\right)/2\right],$$
with
(52)$$\lambda_{3,4}^{DM}=\sqrt{\frac{1}{2}\left[C_{hom}^{DM}\pm \sqrt{{\left(C_{hom}^{DM}\right)}^{2}-4D_{hom}^{DM}}\right]}.$$

Here, $C_{hom}^{DM}$ and $D_{hom}^{DM}$ can be denoted as
(53)$$C_{hom}^{DM}=\frac{A_{DM}\chi_{hom}^{g_{D}}+V\sqrt{B_{DM}}+\eta \left(V+\chi_{line}\right)}{\eta (V+\chi_{tot})},$$
(54)$$D_{hom}^{DM}=\sqrt{B_{DM}}\frac{V+\sqrt{B_{DM}}\chi_{hom}^{g_{D}}}{\eta (V+\chi_{tot})}.$$

In the case of heterodyne detection, the symplectic eigenvalues are expressed as
(55)$$\lambda_{5,6}^{DM}=\sqrt{\frac{1}{2}\left[C_{het}^{DM}\pm \sqrt{{\left(C_{het}^{DM}\right)}^{2}-4D_{het}^{DM}}\right]},$$
and
(56)$$C_{het}^{DM}=\frac{A_{DM}{\left(\chi_{het}^{g_{D}}\right)}^{2}+B_{DM}+1+2\chi_{het}^{g_{D}}\left[V\sqrt{B_{DM}}+\eta \left(V+\chi_{line}\right)\right]+2\eta \left(V^{2}-1\right)}{{\left[\eta \left(V+\chi_{tot}\right)\right]}^{2}},$$
(57)$$D_{het}^{DM}={\left[\frac{V+\sqrt{B_{DM}}\chi_{het}^{g_{D}}}{\eta (V+\chi_{tot})}\right]}^{2}.$$

Finally, we can plot the secret key rate for proposed scheme in terms of discrete modulation.

## 4. Performance Analysis and Results Discussion

In the process of parameter setting of the hybrid amplifier, an appropriate gain value of the amplifier will directly determine the final performance of the scheme. As the main function of DLA is to improve the functional uncertainty caused by NLA, its gain value does not significant improvement for the performance of proposed scheme. Therefore, we mainly discuss the optimal range of NLA gain first. As shown in Figure 5, Figure 6 and Figure 7, here we set the tunable distance to derive the optimal range for $g_{N}$ in our proposed scheme. The parameters are set as fixed value with $g_{D}=12$, η=1, ε=0.01, and $V_{M}=0.3$.

In the first scenario, we set different distance with *d* = 15 km, *d* = 20 km, and *d* = 25 km. As shown in Figure 5, the public zone 1 highlighted with light gray denotes that two kinds of proposed scheme can obtain relative higher secret key rate in this gain value range of $g_{N}$$\left(6.1\leq g_{N}\leq 8.6\right)$.

In the second scenario, we have different distance values with *d* = 30 km, *d* = 35 km, *d* = 40 km, and *d* = 45 km. The public zone 2 has been highlighted with light gray in Figure 6, which represents our proposed schemes can obtain relative higher secret key rate in this gain range with NLA $\left(6\leq g_{N}\leq 14\right)$.

In the final scenario, the distance value are given as *d* = 90 km, *d* = 95 km, and *d* = 100 km. Our proposed scheme can reach the relative higher secret key rate in the public zone 3, which has been shown in Figure 7 with the gain value range $\left(7.5\leq g_{N}\leq 20\right)$.

Therefore, considering the above three scenarios, we can get the overlapping area $\left(7.5\leq g_{N}\leq 8.6\right)$ in Figure 8, which represents the optimal gain range for proposed scheme.

Moreover, we find that the distance between Alice and Eve ($L_{1}$) can also influence the performance of our scheme. In Figure 9, here we set different distances, $L_{1}$, to observe the schemes performance with $L_{1}$ = 0.2 km, $L_{1}$ = 0.3 km, $L_{1}$ = 0.4 km, $L_{1}$ = 0.5 km, and $L_{1}$ = 0.6 km. The fixed parameters are set with $g_{N}=8$, $g_{D}=12$, η=1, ε=0.01, and $V_{M}=0.3$. As shown in Figure 9, the light blue solid lines and red solid lines represent four-state discrete modulation with untested source via heterodyne detection and homodyne detection, respectively. Furthermore, the dark blue solid lines and orange solid lines denote four-state discrete modulation with untested source via hybrid amplifier by heterodyne and homodyne detection, respectively. We can find that with the increasing of $L_{1}$ the maximum transmission distance of all schemes is increasing, and the increasing of scheme with four-state untested source via homodyne detection is the most obvious, which is almost catching up the scheme with four-state untested source by hybrid amplifier via heterodyne detection. However, when the distance exceeds 0.6 ($L_{1}$ > 0.6 km), the four-state with untested source schemes are on longer to generate the secret key rate, and it has the same situation with the counterpart schemes of hybrid amplifier in $L_{1}$ > 50 km and $L_{1}$ > 2 km, respectively (as shown in Figure 10 and Figure 11). Therefore, in our scheme we set the distance between Alice and Eve with $L_{1}=0.5$.

Based on the calculation process in the previous section, we show the performances of various protocols over the lossy quantum channel. According to different modulation methods, we divide these protocols into two categories that includes the protocol based on Gaussian modulation and discrete modulation. For these schemes, we set the entanglement source in untested side, and implement a hybrid amplifier at Bob side. In the case of Gaussian modulation we have the squeezed-state with homodyne and heterodyne detection, or, equivalently, coherent-state with homodyne and heterodyne detection (in Figure 12a). In the scenario of discrete modulation, we select coherent-state with homodyne and heterodyne detection (in Figure 12b). The dashed lines (in Figure 12a) indicate the scheme without hybrid amplifier, and the solid lines denote the scheme implemented hybrid amplifier. Where the red dashed line indicates that the scheme by coherent state with untested source and heterodyne detection is far behind its corresponding scheme with hybrid amplifier, and the red solid line, in terms of transmission distance by 36 km. Furthermore, although the scheme of coherent sate with untested source via hybrid amplifier homodyne detection (the green solid line) has lower initial secret key rate than its counterpart (the green dashed line), its transmission distance has been increased 7 km compared to the same scheme without hybrid amplifier (the green dashed line). The squeezed sate with untested source via hybrid amplifier schemes are indicated as a dark blue solid line and light blue solid line, respectively; here, the scheme with homodyne detection (the light blue solid line) reaches the maximal transmission distance of approximately 45 km, and the scheme of heterodyne detection (the dark blue solid line) reaches it in 17 km. Therefore, in the case of the scheme adopting traditional Gaussian modulation, the hybrid amplifier is helpful in improving the transmission distance indeed.

Next, Figure 12b will show the performance result for Gaussian modulation and discrete modulation schemes. In Figure 12b, the dot-dashed lines represent the discrete modulation with untested source via hybrid amplifier, corresponding to the original schemes that did not adopt the hybrid amplifier, marked with red dashed line and yellow solid line, respectively. The remaining schemes are the Gaussian modulation with untested source and hybrid amplifier. We can see the detailed relationship between each line of the scheme, and find that the scheme of classical four-state modulation with untested source heterodyne detection has poorer performance than the same scheme with Gaussian modulation adopted hybrid amplifier, in terms of transmission distance (20 km behind) and initial secret key rate (1.83 bit per pulse behind). However, after being amplified by the hybrid amplifier, the four-state modulation with heterodyne detection scheme, the red dot-dashed line, has better transmission distance performance by 72 km ahead than the Gaussian modulation (red solid line) and 92 km ahead than the unamplified scheme (red dashed line). The scheme of four-state modulation untested source via hybrid amplifier and homodyne detection, the yellow dot-dashed line, reaches the maximal transmission distance with over 146 km. It is 68 km beyond the scheme of discrete modulation unadopted hybrid amplifier (the yellow solid line), and 128 km ahead the scheme of Gaussian modulation with hybrid amplifier. All the simulation results show that even if the entangled source is untested, the discrete modulation with four-state modulation has better performance in key rate and transmission distance than the scheme with Gaussian modulation.

## 5. Conclusions

In this paper, we proposed a scheme of four-state discrete modulation with hybrid amplifier to improve the performance of traditional Gaussian modulation CVQKD while the entangle source is untested. The hybrid amplifier is composed of a MB-NLA and a DLA in series. Due to deployment of the MB-NLA, the noise penalty caused by the detector in the Bob side can be compensated. Moreover, the adoption of DLA compensated the problem of SNR degeneration and amplification probability caused by NLA. The numerical simulations show that our proposed scheme can improve the maximal transmission distance in the four-state modulation CVQKD and Gaussian modulation CVQKD for both homodyne and heterodyne detection, where the entangled source is untested. In terms of possible future research, it could be focusing to implement the hybrid amplifier in Alice and Bob side.

## Figures and Tables

**Figure 1 entropy-22-00882-f001:**
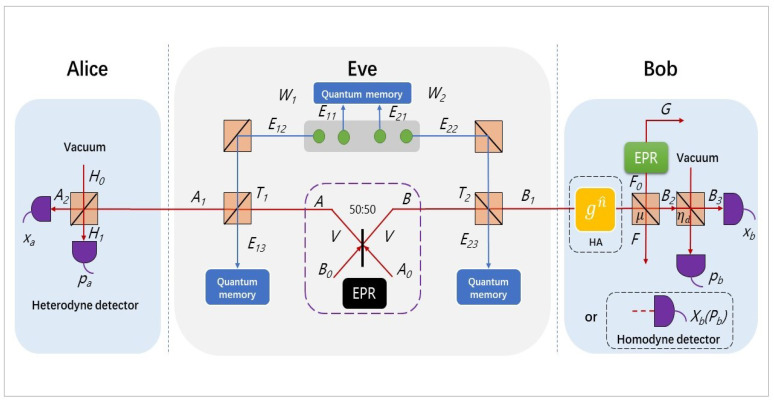
(Color online) Schematic of the entangled source in middle (ESIM) CVQKD using a hybrid amplifier. In the entanglement-based model, Alice detects one of the EPR states by heterodyne detector and the hybrid linear amplifier is installed before Bob uses either the homodyne or heterodyne detector to measure the other half of EPR states. Eve’s attack consists of two entangling cloner attacks on either side of the source. The yellow box of $g^{\hat{n}}$ shows the hybrid linear amplifier.

**Figure 2 entropy-22-00882-f002:**
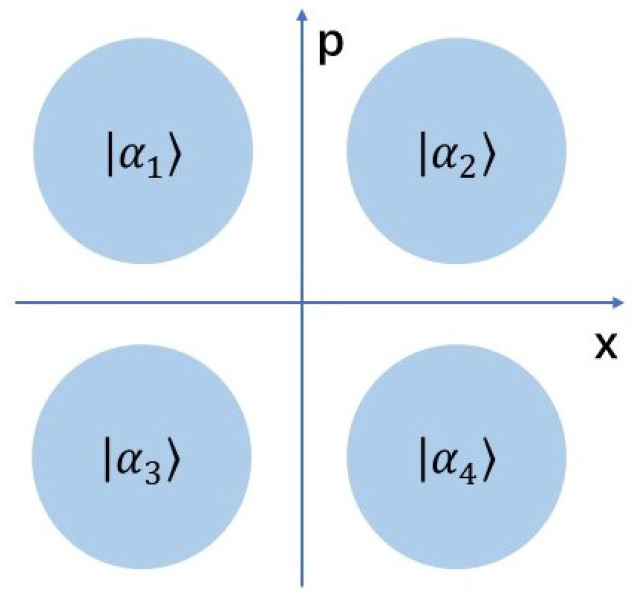
(Color online) Four-state discrete modulation in phase space.

**Figure 3 entropy-22-00882-f003:**
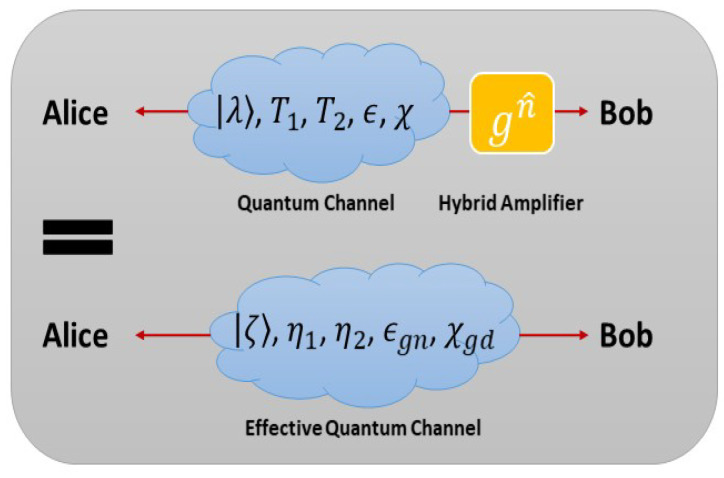
(Color online) An EPR state |λ〉 sent through a Gaussian quantum channel with transmittance $T_{1}$, $T_{2}$, excess noise ϵ, and detection-added noise χ has been replaced by an EPR state |ζ〉 sent through a Gaussian quantum channel with transmittance $\eta_{a}$, $\eta_{2}$, excess noise $\epsilon_{gn}$, and detection-added noise $\chi_{gd}$, but without the hybrid amplifier.

**Figure 4 entropy-22-00882-f004:**
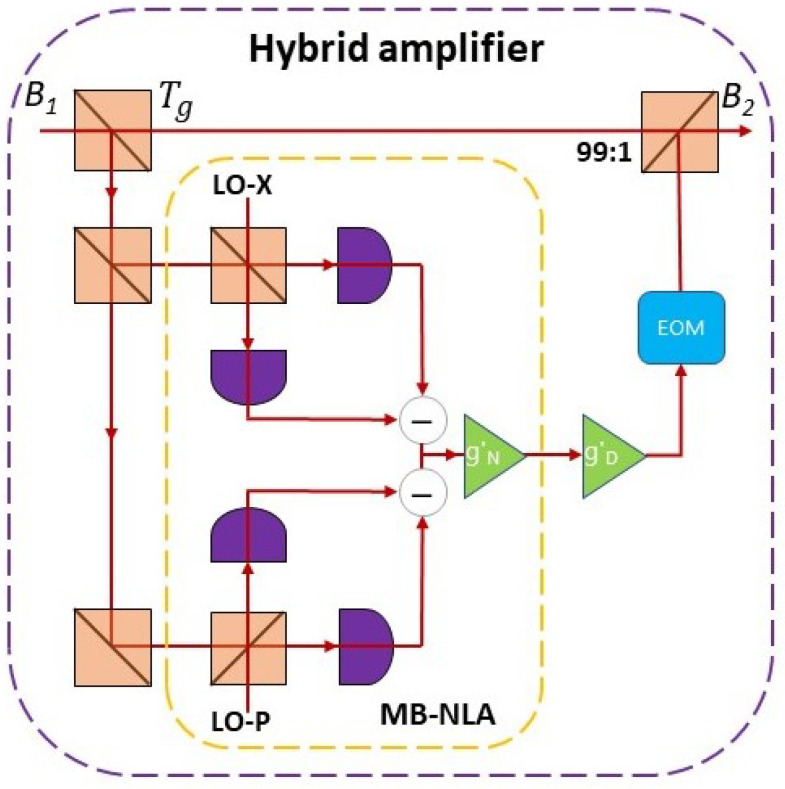
(Color online) The heralded hybrid linear amplifier is applied at Bob side. The mode $B_{1}$ first goes through into a beam splitter with transmissivity $T_{g}$, then the reflected mode goes through into the MB-NLA concatenated by a dual-heterodyne detection and NLA. Here, the dual-heterodyne detection is used to measure the *X* and *P* quadrature of the reflected mode, respectively. After that we set a DLA and an elector-optic modulator (EOM) to dispose the amplified signal pulse and output mode $B_{2}$ by a beam splitter with transmissivity 99:1.

**Figure 5 entropy-22-00882-f005:**
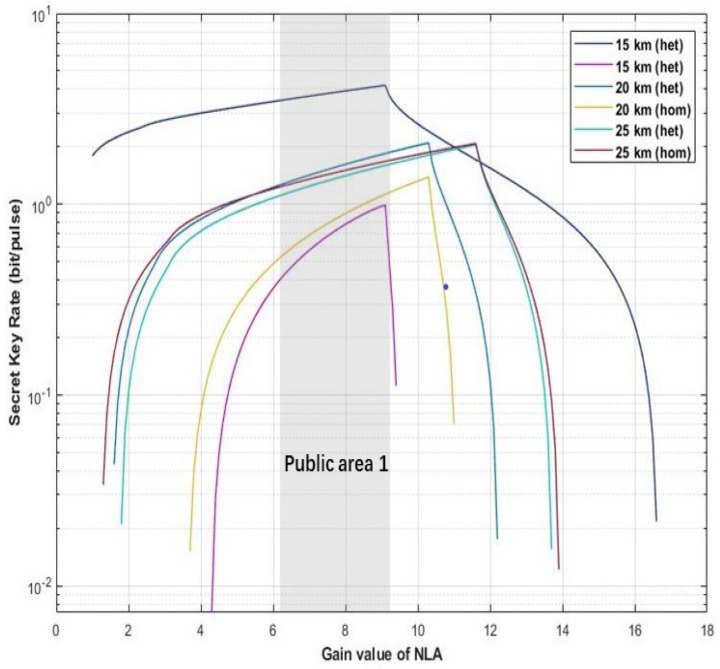
(Color online) The optimal value range for $g_{NLA}$. The X-coordination and Y-coordination represent gain value of NLA and secret key rate, respectively. The figure shows that the proposed scheme with different detector (homodyne detector and heterodyne detector) can obtain relative higher secrete key rate in the public area 1. Here, the distance be set as 15, 20, and 25 km. Moreover, the fixed parameter values are set with $g_{D}=12$, η=1, ε=0.01, and $V_{M}=0.3$.

**Figure 6 entropy-22-00882-f006:**
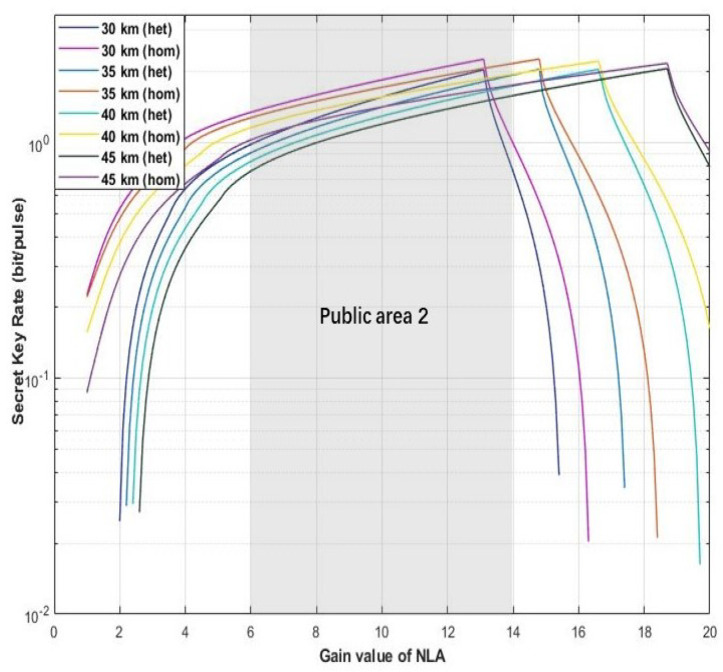
(Color online) The optimal value range for $g_{NLA}$. The X-coordination and Y-coordination represent the gain value of NLA and secret key rate, respectively. The figure shows that the proposed scheme with different detectors (homodyne detector and heterodyne detector) can obtain relative higher secrete key rate in the public area 2. Here, the distance be set as 30, 35, 40, and 45 km. Moreover, the fixed parameter values are set with $g_{D}=12$, η=1, ε=0.01, and $V_{M}=0.3$.

**Figure 7 entropy-22-00882-f007:**
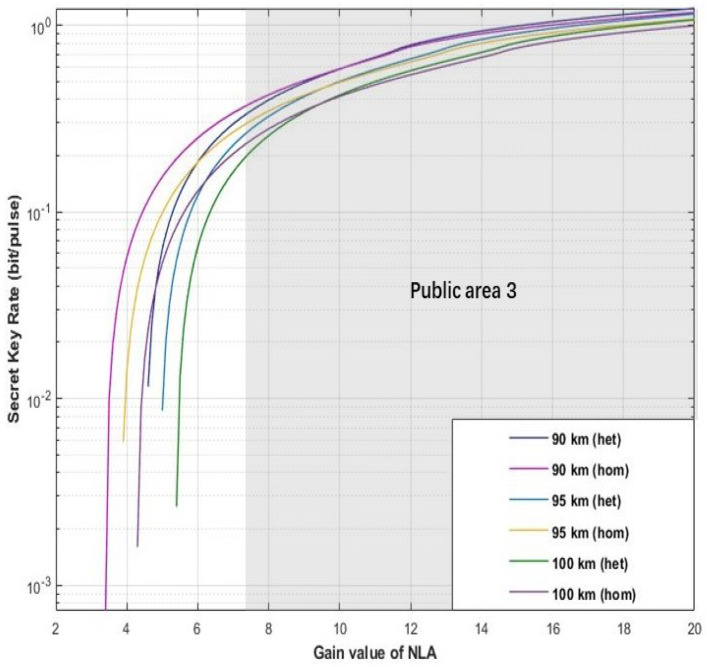
(Color online) The optimal value range for $g_{NLA}$. The X-coordination and Y-coordination represent gain value of NLA and secret key rate, respectively. The figure shows that the proposed scheme with different detector (homodyne detector and heterodyne detector) can obtain relative higher secrete key rate in the public area 3. Here, the distance be set as 90, 95, and 100 km. Moreover, the fixed parameter values are set with $g_{D}=12$, η=1, ε=0.01, and $V_{M}=0.3$.

**Figure 8 entropy-22-00882-f008:**
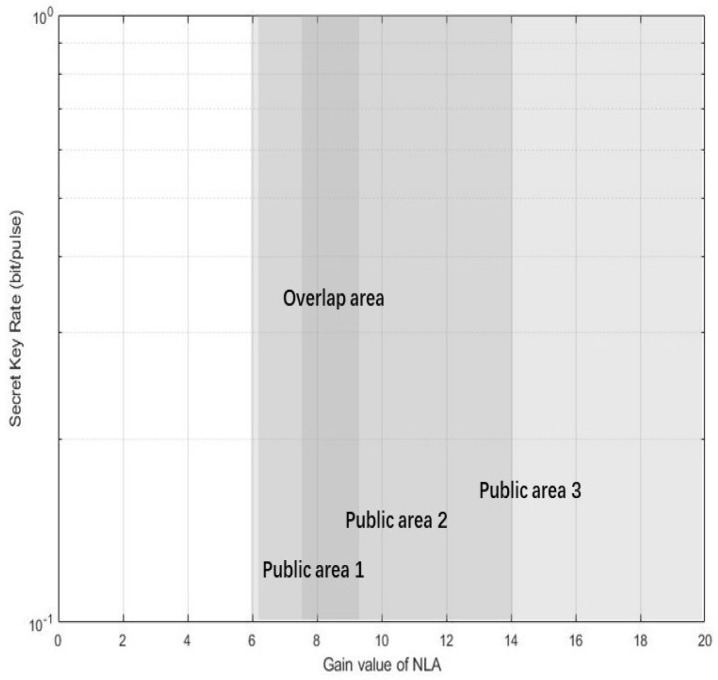
(Color online) Considering the above three public areas denoted in Figure 5, Figure 6 and Figure 7, we get the overlapping region in this figure, which represents the optimal gain value range of proposed scheme.

**Figure 9 entropy-22-00882-f009:**
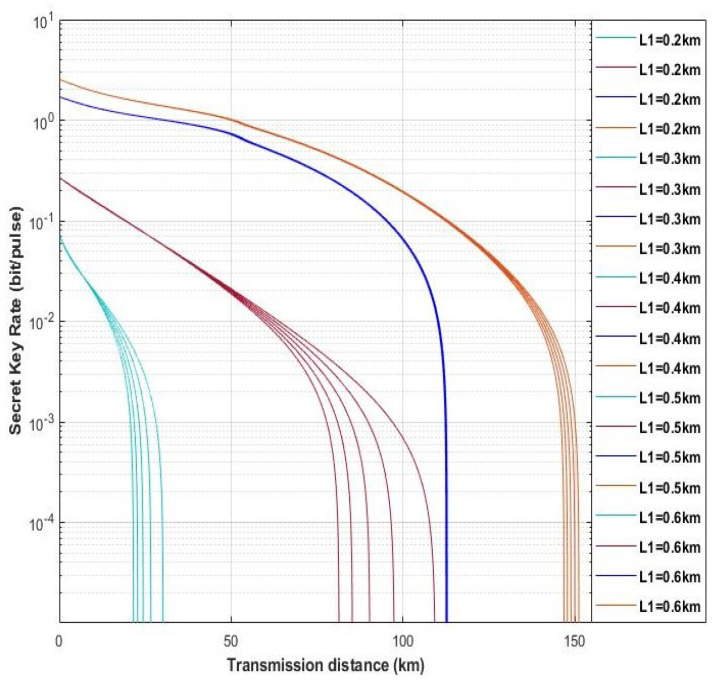
(Color online) The performance of proposed schemes for different $L_{1}$. The fixed parameters are set with $g_{N}=8$, $g_{D}=12$, η=1, ε=0.01, and $V_{M}=0.3$. Here, the light blue solid lines and red solid lines express the scheme without hybrid amplifier, moreover dark blue solid lines and orange lines represent the scheme with hybrid amplifier.

**Figure 10 entropy-22-00882-f010:**
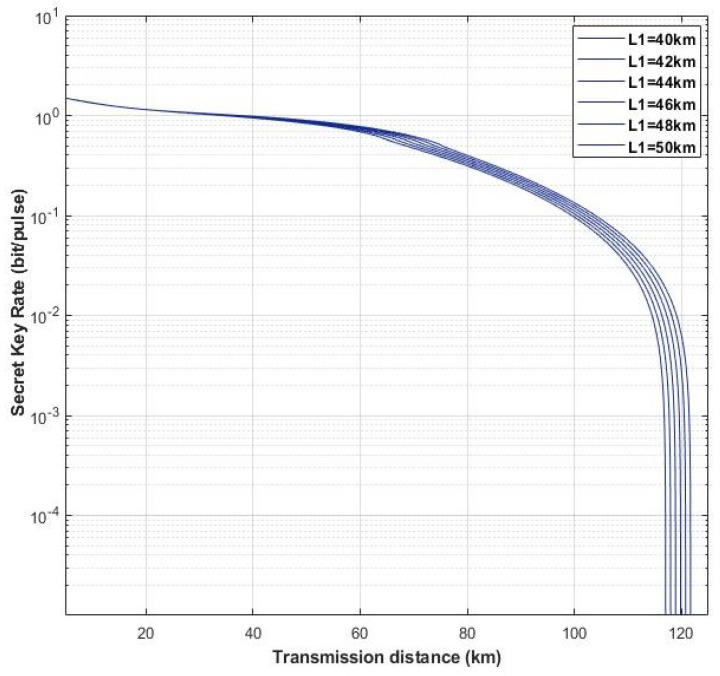
(Color online) The performance of schemes with four-state untested source via hybrid amplifier and heterodyne detection in different $L_{1}$. The fixed parameters are set with $g_{N}$ = 8, $g_{D}=12$, η=1, ε=0.01, and $V_{M}=0.3$.

**Figure 11 entropy-22-00882-f011:**
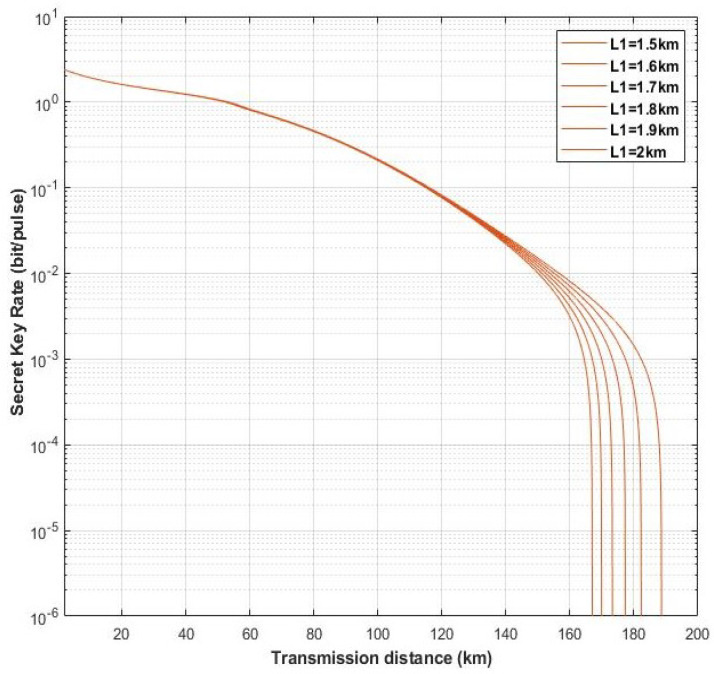
(Color online) The performance of schemes with four-state untested source via hybrid amplifier and homodyne detection in different $L_{1}$. The fixed parameters are set with $g_{N}=8$$g_{D}=12$, η=1, ε=0.01, and $V_{M}=0.3$.

**Figure 12 entropy-22-00882-f012:**
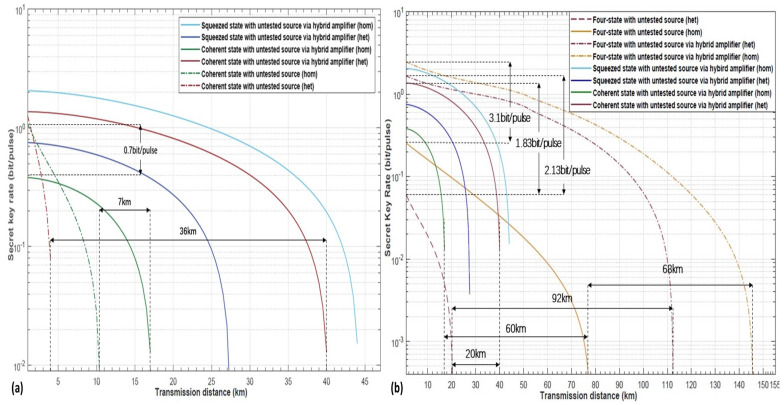
(Color online) Panel (**a**) shows the relationship between the transmission distance and secret key rate. It demonstrates the maximal transmission distance for the scheme with Gaussian modulation. Here, the parameters are set as $g_{N}=8$, $g_{D}=12$, η=1, ε=0.01, and $V_{M}=0.3$. The dot-dash lines in the figure represent Gaussian modulation with an untested source. Furthermore, the solid lines represent Gaussian modulation with untested source via hybrid amplifier. Panel (**b**) also shows the relationship between transmission distance and secret key rate. Here, the parameters are also set as $g_{N}=8$, $g_{D}=12$, η=1, ε=0.01, and $V_{M}=0.3$. In panel (**b**), the green, dark blue, red, and light blue solid lines represent the Gaussian modulation with untested source via hybrid amplifier. Furthermore, the red dash line and yellow solid line represent the four-state discrete modulation with untested source. The red dot-dash line and yellow dot-dash line denote our proposed scheme (four-state discrete modulation with untested source via hybrid amplifier).

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
