# Peer review of "Performance Improvement of Discretely Modulated Continuous-Variable Quantum Key Distribution with Untrusted Source via Heralded Hybrid Linear Amplifier"

_entropy, 2020, doi:10.3390/e22080882_

Round 1
Reviewer 1 Report
The authors improve the performance of the continuous-variable quantum key
by using thefour-state discrete modulation and a heralded hybrid linear amplifier. The authors carry a calculation showing this operation somewhat increases the transmission distance. The paper is interesting and can be published.
Reviewer 2 Report
The present manuscript describes the adoption of a hybrid linear amplifier to improve the performances of CV-QKD.
In my opinion, the paper is not easy to read, and it is difficult to follow the reasoning leading to the claims of the manuscript. A first issue is that several ingredients used during the manuscript (to name a few, the GG02 protocol, Gaussian modulation, a more detailed description of the different amplifiers) are directly mentioned without being introduced. This means that a reader who would like to follow the different steps of the paper has to continuously refer to other papers.
Another issue, at least to my understanding, is exactly what kind of input source the authors are considering. More specifically, in Sec. 2.1 the authors first introduced the four-state coherent encoding, and then in Sec. 2.2 they jump to EPR states. The authors should for instance discuss the connection between
Finally, I would appreciate some intuition on why such approach works. More specifically, when using a NLA, amplification of the signal is related to introduction of post-section (and thus one pays the price in the amount of received signal). In PIA and PSA, the amplification does not introduce post-selection but introduces noise in the amplification process (thus paying a price in such sense). Hence, could the authors provide an intuition on why the proposed Hybrid Amplifier can provide an advantage?
For those reasons, such paper requires extensive re-writing before it can be considered for publication in Entropy.
Round 2
Reviewer 2 Report
The authors have performed some revisions on the manuscript according to my previous report. In the new version of the manuscript the main issues have been clarified, in particular those regarding the source and the advantage provided by the hybrid amplifier.
Before I can recommend its publication in Entropy, I would suggest the authors to perform further revisions to fix some remaining typos and English corrections.
As a minor comment, there is a misprint in Eq. (2) since it does not correspond to a density matrix (the last ket should be changed to a bra).